# Global mortality of snakebite envenoming between 1990 and 2019

GBD 2019 Snakebite Envenomation Collaborators*

Snakebite envenoming is an important cause of preventable death. The World Health Organization (WHO) set a goal to halve snakebite mortality by 2030. We used verbal autopsy and vital registration data to model the proportion of venomous animal deaths due to snakes by location, age, year, and sex, and applied these proportions to venomous animal contact mortality estimates from the Global Burden of Disease 2019 study. In 2019, 63,400 people (95% uncertainty interval 38,900–78,600) died globally from snakebites, which was equal to an age-standardized mortality rate (ASMR) of 0.8 deaths (0.5–1.0) per 100,000 and represents a 36% (2−49) decrease in ASMR since 1990. India had the greatest number of deaths in 2019, equal to an ASMR of 4.0 per 100,000 (2.3−5.0). We forecast mortality will continue to decline, but not sufficiently to meet WHO's goals. Improved data collection should be prioritized to help target interventions, improve burden estimation, and monitor progress.

Snakebite envenoming affects millions of people worldwide annually and is a significant source of mortality[1]. Preventing and treating the problem is complex and requires collaboration among the fields of public health, medicine, ecology, and laboratory science. After being removed from the category A neglected tropical disease (NTD) list in 2013, snakebite envenoming was reinstated in 2017 in response to antivenom shortages and advocacy from researchers and international NGOs[2,3]. In 2019, the World Health Organization (WHO) set a target to halve the number of deaths and cases of snakebite envenoming by 2030[4].

Few studies on the global disease burden of snakebite envenoming have been conducted. In 1998, Chippaux estimated over 100,000 deaths were caused by snakebite envenoming[5]. In 2008, Kasturiratne and colleagues used the Global Burden of Diseases, Injuries, and Risk Factors Study (GBD) framework to capture regional trends and found that snakebite envenoming caused between 20,000 and 94,000 annual deaths globally[6]. At the regional level, meta-analyses have analyzed national health reporting systems, hospital records, and household surveys to estimate the regional burden in sub-Saharan Africa and the Americas, and found the annual mortality to be 7331 and 370 deaths, resepectively[7,8]. Recent community-based household surveys have demonstrated the capacity of targeted data collection to assess the burden of snakebite envenomation in areas of high snakebite vulnerability, such as India, Sri Lanka, and the Terai Region of Nepal[9–13]. Updated estimates of the global situation, including the use of large global health data repositories and more advanced spatiotemporal modeling, are lacking[14].

Here, we present annual estimates of the mortality and years of life lost (YLLs) due to snakebite envenoming in 204 countries and territories from 1990 to 2019 by age and sex using verbal autopsy (VA) survey and official vital registration (VR) mortality data from the GBD cause of death data repository. We discuss our results in light of the WHO goal of halving the number of deaths and cases of snakebite envenoming by 2030 by forecasting the disease burden to 2050. To guide specific public health interventions, we quantify the association between snakebite envenoming and select covariates to better understand what factors are associated with death from snakebite envenoming. We find that the majority of deaths from snakebite envenoming occurred in South Asia, with sub-Saharan Africa having the second-most deaths. Mortality from snakebite envenoming has decreased over the last 30 years, however, at an insufficient rate to meet the WHO's 2030 goal.

## Results

### Global mortality and years of life lost

Snakebite envenoming accounted for 63,400 deaths (95% uncertainty interval [UI] 38,900–78,600) and 2.94 million YLLs (1.79 million−3.74 million) in 2019, globally. This was equal to an age-

standardized rate of 0.8 deaths (0.5–1.0) per 100,000 and 38 YLLs (23–49) per 100,000. From 1990 to 2019, the global age-standardized rate of death and YLLs per 100,000 decreased significantly by 36% (2–49) and 40% (6–55), respectively. Globally, the mortality from snakebite envenoming was greater in males than females in 2019, although non-significantly, with an age-standardized rate of 0.9 deaths (0.6–1.1) per 100,000 in males, compared to 0.7 deaths (0.3–1.0) per 100,000 in females (Fig. 1).

## Burden by region and Socio-demographic Index

Mortality due to snakebite envenoming showed substantial regional variation (Fig. 2 and Supplementary Fig. 5). South Asia had the greatest burden, with 54,600 deaths (95% UI 31,800–68,300) and 2.54 million YLLs (1.48 million–3.21 million), accounting for 86% (76–92) of global deaths and 86% (78–91) of global YLLs (see Supplementary Data file). The age-standardized death and YLL rates were equal to 3.4 deaths (2.0–4.2) per 100,000 and 144 YLLs (83–182) per 100,000, respectively. Western, Central, and Eastern sub-Saharan Africa had the next-highest mortality from snakebite envenoming, with 1.4 deaths (1.0–2.1), 1.3 deaths (0.8–1.8), and 1.2 deaths (0.8–1.6) per 100,000, respectively. The regions with the lowest age-standardized rates in 2019 were Central Europe, high-income North America, high-income Asia Pacific, and Western Europe. At the regional level, there was a log-linear relationship between the Socio-demographic Index (SDI) of a region and the region's age-standardized snakebite envenoming mortality rate in 2019 (Fig. 3).

India had the greatest absolute number of snakebite envenoming deaths in 2019 at 51,100 deaths (95% UI 29,600–64,100), followed by Pakistan (2070 deaths [1470–2950]). In India, the age-standardized rate of death due to snakebite envenoming was 7.3 per 100,000 (4.1–8.8) in 1990 and decreased to 4.0 per 100,000 (2.3–5.0) in 2019, which represents the greatest absolute decrease over that timespan globally. Within India, Chhattisgarh, Uttar Pradesh, and Rajasthan had the greatest age-standardized death rates, at 6.5 deaths (3.5–8.4), 6.0 deaths (2.6–8.0), and 5.8 deaths (3.5–7.4) per 100,000, respectively. Uttar Pradesh had the greatest absolute number of deaths of any state in India in 2019, with 12,000 deaths (5230–16,100). See Supplementary Table 5 for state-level results for all of India.

Sub-Saharan Africa also had a high burden of snakebite envenoming deaths across the entire continent. Nigeria had the greatest number of deaths with 1460 (977–2640), and there were seven other countries in Western sub-Saharan Africa with greater than 200 deaths (Burkina Faso, Cameroon, Chad, Côte d'Ivoire, Ghana, Mali, and Niger). In Central sub-Saharan Africa, the Democratic Republic of Congo had 545 deaths (313–1030). In Eastern sub-Saharan Africa, Ethiopia had 499 deaths (321–708) and Kenya had 349 (197–603). Somalia, Central African Republic, and Eritrea had the greatest age-standardized death rates in sub-Saharan Africa, at 4.5 (1.6–14.1), 3.4 (2.1–5.6), and 2.9 (1.2–5.2) deaths per 100,000, respectively.

## Forecasted mortality to 2050

By 2050, the rate of snakebite envenoming mortality globally is expected to decrease to an age-standardized rate of 0.7 deaths (95% UI 0.4–1.1) per 100,000 (Fig. 4). This is equivalent to 68,800 absolute deaths annually (39,100–126,000), which is greater than the number of deaths that occurred in 2019, due to forecasted population increases. By 2030, we predict the global age-standardized rate will non-significantly decrease by 8.6% (−9.6 to 20.1). See Supplementary Table 6 for each region's forecasting results, by decade, from 2020 to 2050.

## Discussion

Snakebite envenoming caused 63,400 deaths (95% UI 38,900–78,600) and 2.9 million YLLs (1.8 million–3.7 million) in 2019, which makes it the deadliest NTD according to GBD 2019[15]. Over time, the global age-standardized rate of death has decreased by 36% (2–49), which shows progress; however, this annual rate of change would be insufficient to accomplish WHO's 2019 goal of halving the burden by 2030[4].

South Asia had the greatest mortality from snakebite envenoming due to the intersection of ecological factors, socioeconomic vulnerability, and low health system capacity, which creates a population at risk of snakebite envenomation and death. Specifically, India had the greatest number of deaths, with over 50,000 in 2019. These estimates are consistent with previous research conducted with verbal autopsy mortality surveys, which were the source of data in India in our analysis as well[9–11]. After a venomous snakebite occurs, the probability of death

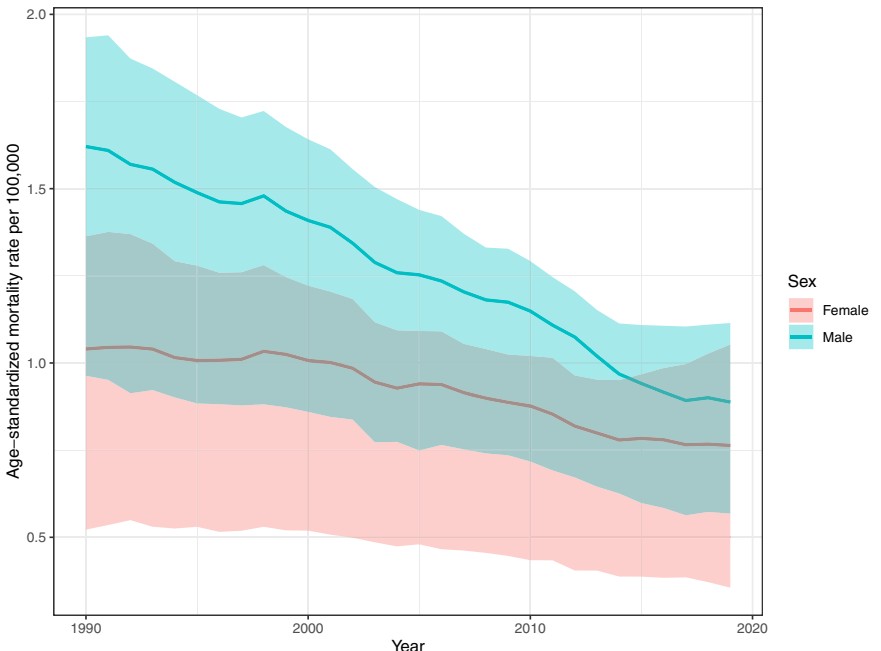

**Fig. 1 | Global age-standardized mortality rate of snakebite envenoming in males and females from 1990 to 2019.** The upper and lower estimates of the 95% uncertainty interval are represented by the error bands around the mean estimate. Source data are provided as a Source Data file.

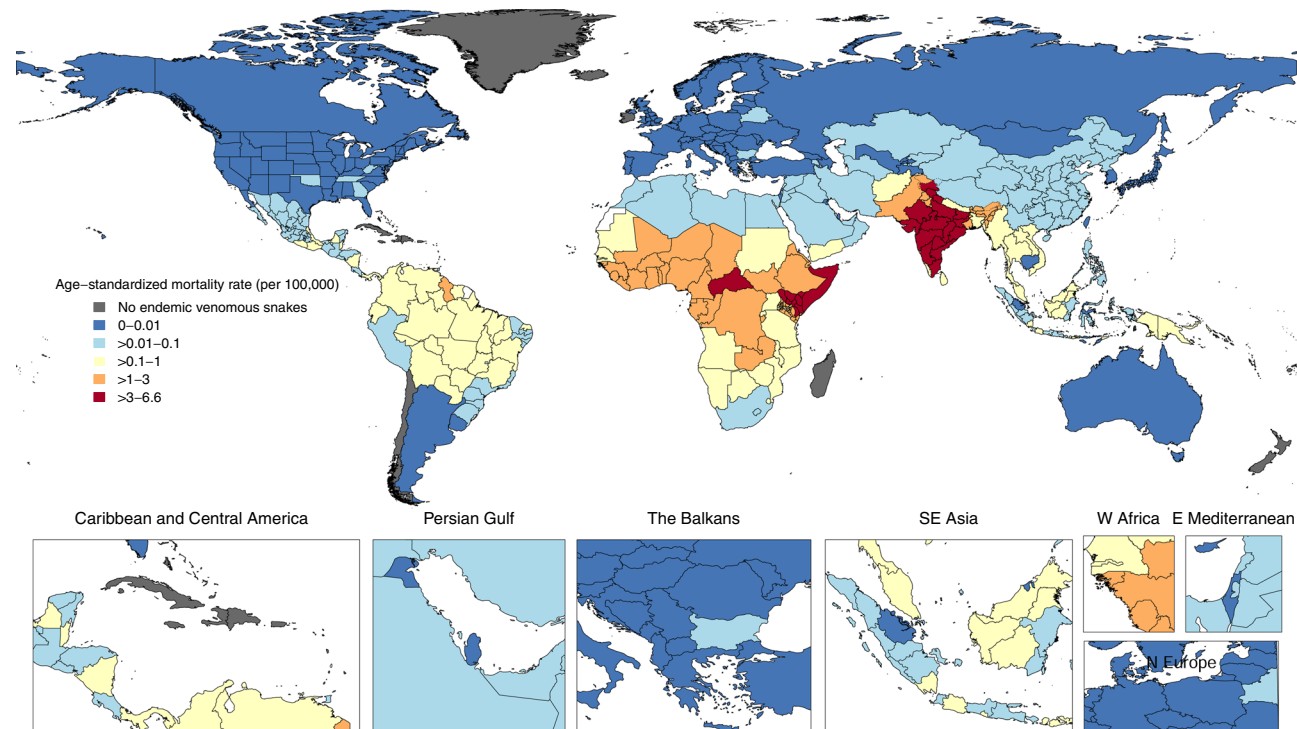

**Fig. 2 | Age-standardized mortality rate of snakebite envenoming in 2019 across 204 countries and territories.** Age-standardized snakebite envenoming mortality rates across both sexes combined in 2019. GBD 2019 did not publish state-level estimates for China, so each state is colored based on China's national estimate. The endemic habitat of venomous snakes of medical importance was queried from the WHO Snakebite Information and Data Platform[31]. Source data are provided as a Source Data file.

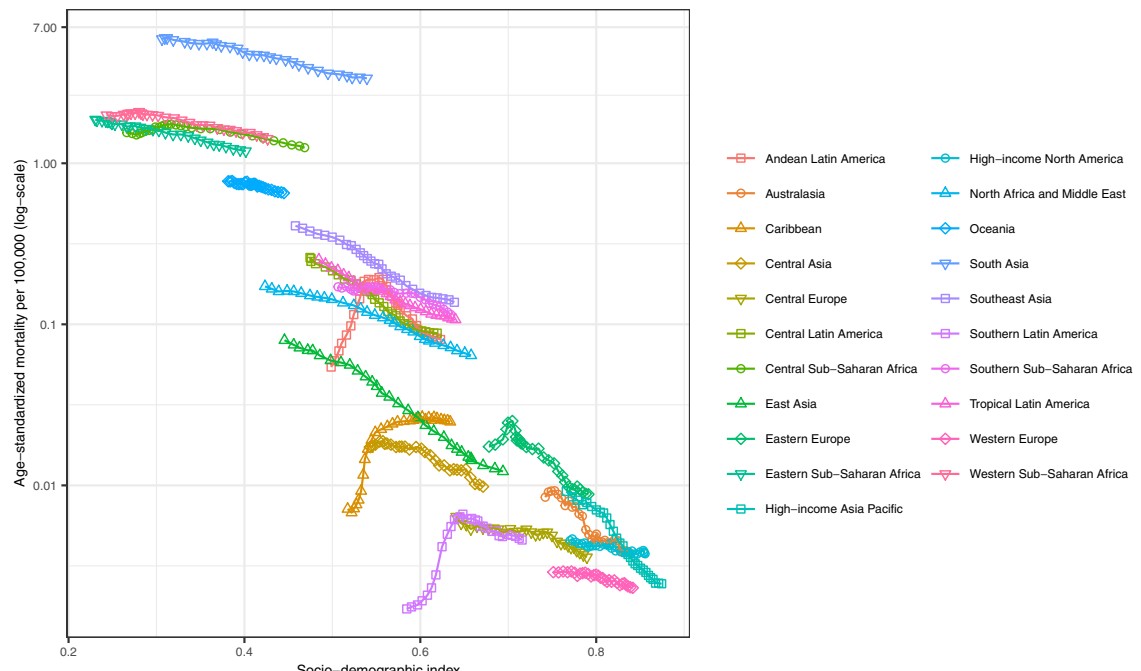

**Fig. 3 | Regional age-standardized mortality rate from snakebite envenoming from 1990 to 2019 by Socio-demographic Index value of the region.** Age-standardized snakebite envenoming mortality rate per 100,000 by region and Socio-demographic Index. Each point represents the age-standardized mortality in a given year from 1990 to 2019 in the region. Y-axis is on log scale. Source data are provided as a Source Data file.

increases if antivenom is not administered within six hours[16]. However, in South Asia, many seek out traditional healers or attend clinics with insufficient education about how to treat snakebite envenoming or lacking the antivenom to administer life-saving treatment[16–19]. Victims who do reach a hospital often have insufficient access to dialysis,

ventilators, and blood transfusions, which are essential to deal with the complications of envenoming[18,20]. Interventions to secure more rapid antivenom delivery need to be coupled with preventive strategies like increased education and health system strengthening in rural areas, and be targeted around the geographical and seasonal variation of

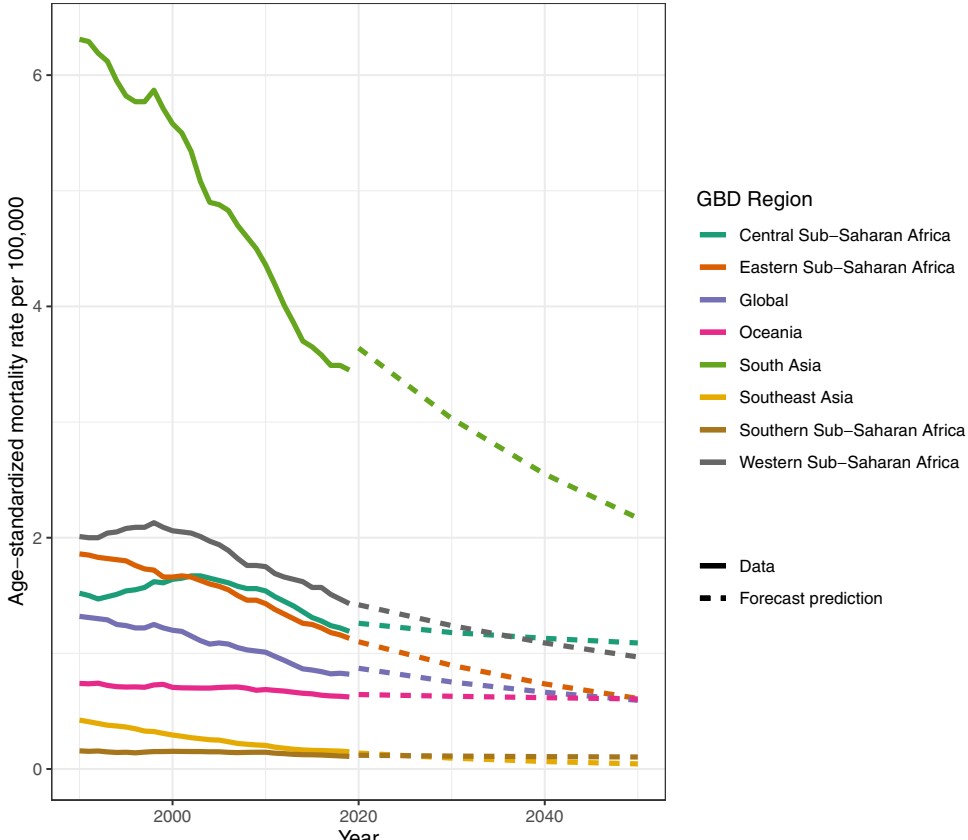

**Fig. 4 | Global and regional age-standardized snakebite envenoming mortality rate from 1990 to 2019 with forecasting to 2050 of the seven regions with the highest snakebite envenoming burden.** Age-standardized snakebite envenoming mortality rate per 100,000 by region and year, forecasted to 2050. The plot shows the top seven regions in terms of age-standardized rates in 2019, all of which had age-standardized mortality rates greater than 0.1 per 100,000. Lines in bold are the snakebite envenoming estimates from the primary statistical analysis pipeline, while dotted lines are the predictions from the forecast regression. No steps were made to align GBD 2019 cause-specific mortality rate estimates with the predicted forecast from 2020, and predictions are made based on the average annualized rate of change from 1990 to 2019 and the age-sex demographic composition of each region. Source data are provided as a Source Data file.

snakebite envenomation risk to maximize their ability to prevent and treat snakebites.

Sub-Saharan Africa faces many of the same problems as South Asia, such as health system capacity shortcomings and the use of traditional healers as primary providers, as well as problems like an inadequate production of antivenom for the continent's endemic snakes and high rates of conflict and humanitarian crisis[21,22]. We estimated Sub-Saharan Africa had the second greatest mortality with 6790 deaths (95% UI 5040–10,100) and 314,000 YLLs (219,000–521,000), equivalent to age-standardized rates of 1.2 deaths (0.9–1.6) per 100,000 and 36.9 YLLs (27.3–54.6) per 100,000. The supply of antivenom is inadequate in sub-Saharan Africa and the cost for a dose is often prohibitively expensive, leaving victims of snakebite envenomation without treatment options[22]. Many of the countries with the highest burden of snakebite envenoming in sub-Saharan Africa are also those recently or currently in the midst of conflicts and humanitarian crises that increase outdoor exposure and disrupt the health system's capacity for surveillance and treatment[21,23]. For example, Somalia (4.5 deaths per 100,000), Central African Republic (3.4 deaths per 100,000), Eritrea (2.9 deaths per 100,000), Chad (2.6 per 100,000), and South Sudan (2.3 deaths per 100,000) have some of the highest mortality rates globally. Human displacement from conflict likely leads to increased human-wildlife conflicts and decreased access to antivenom and other necessary medical care[14,21]. Our estimates corroborate previous reports that populations in the middle of humanitarian, migration, and environmental crises are at high risk and surveillance efforts should be scaled up targeting these populations[21].

Our ensemble modeling framework allowed us to test multiple covariates for their association with snakebite envenoming mortality and provided important insights on the disease's epidemiology. Environmental indicators such as living at a lower elevation and latitude and socioeconomic indicators like education had strong negative associations with snakebite envenoming mortality (Supplementary Fig. 4). Education had a more negative association for males, while urbanicity was more strongly negative for females. These findings aligned with previous research that reported higher snakebite envenoming mortality in females than males in rural areas[11]. We show that at a population level, interventions for rural areas focused on antivenom delivery should be supplemented with education for agricultural workers to increase awareness of high-risk behaviors and mitigation strategies. With more granular geospatial and temporal epidemiological data, streamlined and targeted interventions can be achieved, such as the use of education, rapid emergency transport for agricultural workers, antivenom delivery to high-risk areas, and rigorous evaluation of innovative interventions like antivenom delivery via drones to at-risk rural locations[14,24–27].

When paired with the recent analysis by Longbottom et al. that mapped the vulnerability to snakebite envenoming, our estimates present a complementary assessment of the drivers behind snakebite envenoming mortality, and especially highlight where there are gaps in antivenom access across the world[28]. In some places, Longbottom et al.'s results intersected with locations we estimated to have high mortality rates, such as Central and Eastern sub-Saharan Africa, which Longbottom et al. estimated have significant vulnerability due to poor

health system infrastructure and the presence of snakes for which there is no effective antivenom. Conversely, we found that high rates of mortality also occur in areas that Longbottom et al. did not estimate to have a high vulnerability, such as India. This is likely due to the existence of antivenom for the Big Four snakes (*Bungarus caeruleus, Daboia russelii, Echis carinatus,* and *Naja naja*) that cause over 90% of envenomations in the country, while the vulnerability estimates were focused on exposure to snakes that do not have antivenom treatments[16,28]. Our mortality estimates demonstrate that preventing snakebite envenoming death depends on not just the existence of antivenom, but also its dissemination to rural areas and the health system's capacity to provide wound care and necessary medical treatment for victims with secondary complications such as neurotoxic respiratory failure or acute kidney injury requiring dialysis[29].

In this analysis, we incorporated an extensive amount of ICD-coded VR and VA data that have previously not been utilized in global snakebite estimates. However, even in this dataset, there was sparsity across some locations, for example, in sub-Saharan Africa and Southeast Asia, where there are few robust in-country data reporting systems. Our estimate of 6790 deaths (95% UI 5040–10,100) in sub-Saharan Africa aligns closely with the meta-analysis by Chippaux, which estimated there were 7331 (5149−9568) annual deaths[7]. Both studies had similar limitations due to data scarcity, are likely underestimates of the true number of deaths, and emphasize the urgent need for better epidemiological assessments to provide a more accurate estimation of the true disease burden due to snakebite envenoming in high-risk areas like sub-Saharan Africa, South Asia, and Southeast Asia.

VA and VR are both imperfect methods for counting deaths from snakebite envenoming and represent another limitation in our study. Using VA data, we could still be underestimating the true magnitude of death if the distinctive signs of snakebite, or the snake itself, were not seen when the bite occurred. For example, in Cambodia, only a single verbal autopsy study including venomous animal mortality has been conducted to our knowledge[30], which did not find a single death due to snakebite envenoming, despite the presence of multiple venomous snakes in the country[31]. Snakebite envenoming deaths are also rare enough that it is difficult for surveys to identify a sufficient number of deaths to estimate a robust mortality estimate. In Sri Lanka, a recent study by Ediriweera et al. using household surveys in 165,000 people found only five deaths in the whole country, equal to a rate of 2.3 per 100,000 people, with a 95% confidence interval of 0.2 to 4.4[12]. We used nationally representative and complete VR data from Sri Lanka in our estimation process; however, official death statistics have been shown to miss many snakebite envenoming deaths or miscode them as another cause. Studies comparing verbal autopsy community-based studies and official records frequently find that official records undercount the number of deaths that actually occurred[10,11,19]. While the VR data we used in Sri Lanka likely underestimated the true community-level disease burden, Ediriweera et al's study demonstrate the limitations in the ability of verbal autopsy to calculate precise and accurate rates of rare events like snakebite envenomation deaths. Acknowledging the limitations in vital registration data, we attempted to use post-processing steps like redistribution of ill-defined causes of death to attempt to account for underreporting[32]. Given that many snakebite envenoming deaths occur in rural settings in countries without robust surveillance, our estimates are likely underestimates given the limitations of the epidemiological data.

Another limitation in our analysis was that we were only able to present results for each country, while recent verbal autopsy surveys have shown granular geographical and seasonal variation in the risk of snakebite envenomation within countries, with the greatest risk in rural areas[12,13]. One recent study in the Terai region of Nepal, which is a low-altitude zone with a monsoon climate where agriculture is the primary occupation, found that the mortality rate of snakebite

envenomation was 22.4 per 100,000, over five times our estimate for India[13]. According to mortality estimates from GBD 2019, these results would make snakebite envenomation the ninth greatest cause of death in Nepal and the greatest cause of death among injuries, ahead of the mortality rate due to falls, self-harm, and road injuries[15]. More large-scale community-based surveys are required to accurately determine the burden of snakebite envenoming in rural areas and better understand the geographic and temporal trends that could lead to more impactful interventions. For example, in South Asia, snakebite envenomation is closely tied to monsoon season, which should guide health system infrastructure planning and antivenom distribution, among other interventions[13].

To improve future studies, questions related to snakebite envenoming should be incorporated into regular health surveys that are already being conducted across sub-Saharan Africa and South Asia. In WHO's 2019 Strategy for Prevention and Control of Snakebite Envenoming in sub-Saharan Africa, updated and precise epidemiological data were outlined as a need moving forward to better guide appropriate and efficient implementation of antivenom interventions[33]. Injury surveillance, such as the use of District Health Information System 2 (DHIS2), has shown promise and could be adapted to snakebites to create real-time geographically specific epidemiological monitoring[34]. Increased collaboration between researchers and local health institutions should be prioritized to bolster the availability of data, demonstrate the unmet need for antivenom, and rigorously monitor and evaluate interventions.

Our analysis also relied on WHO venomous snake distribution data to decide which locations could reliably be identified as having venomous snakes of medical importance and which did not. It was important for our results to be ecologically feasible, and this database represented the most complete list of venomous snakes capable of causing mortality that we could find. However, while it is updated iteratively, it is not complete and only contains approximately 200 venomous snakes deemed medically important, out of 600 venomous snakes. While these other 400 snakes may not cause fatalities regularly, they could cause fatal envenomation in rare cases. If a country only contained one of these 400 venomous snakes that was capable of a rare fatal envenomation and not one of the 200 medically important snakes, then we would be erroneously zeroing out that location. For example, there is the Solomons Coral Snake (*Salomonelaps par*) in the Solomon Islands that has no recorded fatal envenomations, but there are case reports of near-lethal bites[35]. Conversely, we had official health statistics data that recorded an ICD-coded death due to snakebite envenoming in Chile and New Zealand, but based on the review of the WHO venomous snake distribution database and venomous snake habitats, we agreed that there were no endemic venomous snakes despite these recorded deaths.

In conclusion, we provide the most comprehensive and data-driven estimates of the global magnitude of snakebite envenoming mortality to date. We find that deaths are concentrated in South Asia; however, sub-Saharan Africa also has a high disease burden. Significant investments in data collection, research, and public health intervention are required to better quantify the magnitude of snakebite envenoming. Securing timely antivenom access across rural areas of the world would save thousands of lives, and greater investment into devising and scaling up these interventions should be prioritized to meet WHO's snakebite envenoming and neglected tropical disease goals.

## Methods
### Summary
We started by reviewing GBD 2019 mortality estimates for venomous animal contact. The GBD study and its methodological framework to estimate mortality due to injuries have been described in detail elsewhere[15,36].

In brief, we used a subset of the data for venomous animal contact to identify snakebite-specific mortality, as well as other animal-specific mortality, and evaluated these data using models that captured spatiotemporal patterns to estimate mortality for four different animals (snakes, bees, scorpions, spiders) and for a fifth residual category (other venomous animal contact). We adjusted each animal-specific mortality estimate so that their sum equaled the GBD 2019 overall venomous animal contact mortality estimates, thus preserving internal consistency. To account for uncertainty in the primary data, data processing, measurement error, and choice of model, every model in the process was run 1000 times to produce final estimates with 95% uncertainty intervals, which comprise the 2.5th and 97.5th percentiles of 1000 draws.

## GBD 2019 venomous animal contact estimation

We used published GBD 2019 estimates for overall venomous animal contact mortality as a platform for our analysis. A summary of the GBD 2019 estimation approach for mortality from venomous animal contact follows.

The case definition for a venomous animal contact death in GBD 2019 was death resulting from unintentionally being bitten by, stung by, or exposed to a non-human venomous animal. We identified deaths in VR and VA cause of death data using ICD-9 codes E905-E905.99 and ICD-10 codes X20-X29.9. Once data from all available sources were identified, data underwent the processing that occurs for all cause-of-death data in GBD, which includes noise reduction to reduce stochastic variation and redistribution of unspecified or incorrectly coded causes of death. This is important for snakebite envenoming, which can manifest in multiple injuries after systemic envenomation and be misattributed to a different cause of death. These data preparation steps for GBD mortality estimates are described in detail elsewhere[15]. See Supplementary Fig. 1 for a map of data used in the GBD 2019 venomous animal contact model.

Next, mortality due to snakebite envenoming was modeled using GBD cause of death ensemble modeling (CODEm). CODEm explores a large variety of possible submodels to estimate trends in causes of death using an algorithm to select varying combinations of covariates that are run through several modeling classes[37]. Covariates are also included to guide predictions where data are sparse or absent. In this analysis, we included 16 socioeconomic or environmental covariates identified as conceivably associated with the risk of snakebite mortality risk: (1) population-weighted rainfall in mm/yr[TV], (2) urban proportion of the location[TV], (3) Proportion of population involved in agricultural activities[TV], (4) Population-weighted mean temperature[TV], (5) Absolute value of average latitude, (6) Proportion of the location over 1500 meters elevation, (7) Proportion of the location under 100 meters elevation, (8) Population density over 1000 people per square kilometer (binary)[TV], (9) Population density under 150 people per square kilometer (binary)[TV], (10) Healthcare Access and Quality Index[TV], an estimate from the GBD that describes a country's healthcare access and quality[38], (11) Socio-demographic index[TV15], (12) lag-distributed income per capita[TV], (13) Education in years per capita[TV], (14) Log-transformed Summary Exposure Value for venomous animal contact[TV], a measure of a population's exposure to a risk factor that takes into account the extent of exposure by risk level and the severity of that risk's contribution to disease burden[39], (15) Proportion of population vulnerable to venomous snakebites[28], and (16) Mean number of venomous snake species inhabiting a location[28]. Every covariate was available for all 204 countries. A subset was time-varying (indicated by superscript TV). Each covariate was given a prior on the direction of its beta coefficient, either positive or negative, if there was a strong prior that it was associated in a positive or negative manner with venomous animal contact mortality.

The predictive validity of each of the submodels was tested using test-train holdouts, whereby a specific model was trained on 70% of the data and tested on the withheld 30% of data to determine out-of-sample predictive validity, which was quantified using root mean-squared error (RMSE). Once the submodels were conducted and predictive validity was measured, then an ensemble model was developed out of the submodels. The best-performing models were chosen based on out-of-sample predictive validity.

YLLs are defined as the difference between life expectancy and the age at which a death occurs, based on life tables used in GBD 2019 that estimate the remaining life expectancy for each five-year age group in all populations greater than 5 million in GBD 2019. Supplementary Table 2 shows the life expectancy used in YLL calculations for GBD 2019.

## Study design and data sources

After GBD 2019 venomous animal contact mortality was estimated, we undertook the following steps to estimate snakebite-specific mortality.

We first reviewed all cause of death data that could be mapped directly to snakebites or other venomous animals. The ICD codes used for each animal are listed in Supplementary Table 3, along with the volume and type of data used in snakebite modeling. The snakebite-specific model had 10,636 location-years of data. See Supplementary Figs. 2, 3 for maps of the volume of data used in the snakebite envenoming model and the type of data in each location.

After obtaining all possible data, we applied the same cause of death noise reduction processing described above to the raw animal-specific data[15]. We redistributed deaths coded to ICD codes E905, E905.9, and X29—which code for deaths due to unspecified venomous animals—by aggregating all the properly coded deaths by location, age, sex, and animal and applying the proportion of correctly coded deaths due to snakebites to the number of deaths coded for an unspecified venomous animal. Redistribution was based on location, age, and sex patterns from correctly coded venomous animal deaths. Out of 69,097 deaths that could be coded to the ICD codes above, 5711 (8.3%) were coded to unspecified venomous animal contact and needed to be redistributed. Redistributed animal deaths were added to the number of properly coded deaths for each animal. If a location-age-sex group had more incorrectly coded deaths that needed to be redistributed than properly coded deaths across all five animal groups, then we aggregated based on a broader demographic in order to have a more stable proportion for redistribution. First, we aggregated the codes by only location and age and applied these proportions to the location-age-sex groups where there were sufficiently properly coded deaths by location and age, but not when stratified by sex. If there were still insufficient deaths when disregarding sex, we aggregated across all ages and both sexes within a location and applied that proportion. If there were still more deaths needed for redistribution than properly coded deaths in a location, we aggregated deaths over the GBD region to estimate the proportion of deaths due to each animal and applied that proportion to the redistributed deaths. There were 27,020 deaths properly coded for snakebites. After redistribution, there were 29,040 deaths attributable to snakebites, an increase of 7.5%.

## Statistical analysis

Following noise reduction and redistribution of ill-defined causes of death, we developed statistical models based on the spatiotemporal Gaussian process regression (ST-GPR) modeling framework used in GBD[15]. ST-GPR starts by fitting a mixed-effects linear prior and then fitting a second model based on the weighted residuals between the input data and the linear prior. We set the second-stage model weights to allow high smoothing over time due to a prior expectation that the burden of snakebite does not change substantially year after year, low smoothing over space because of a prior that the burden of countries within a region can vary substantially, and a medium weight over age to allow age smoothing while not overfitting. See *ST-GPR parameters* in the Supplementary Information for further details on ST-GPR

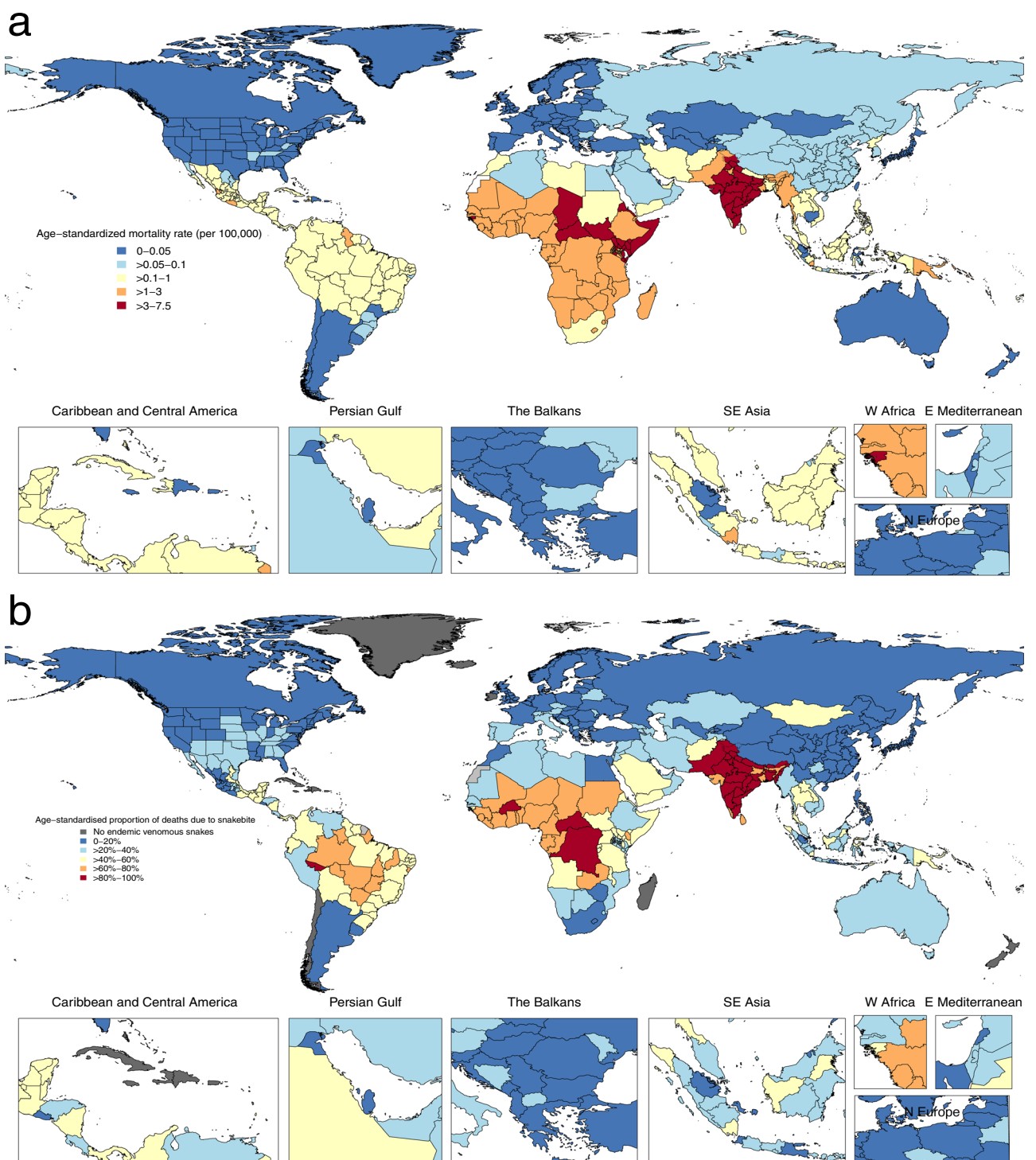

**Fig. 5 | Age-standardized mortality rate from all venomous animal contact and the age-standardized proportion due specifically to snakebite envenoming.** **a** GBD 2019 estimates of the age-standardized mortality rate from venomous animal contact for both sexes combined in 2019. **b** Estimated age-standardized proportion of all venomous animal contact deaths due to only snakebites in 2019. GBD 2019 did not publish state-level estimates for China, and each state is colored based on China's national estimate. The endemic habitats of venomous snakes of medical importance was queried from the WHO Snakebite Information and Data Platform[31].

hyperparameter weighting equations and covariate selection. Every combination of covariates (Supplementary Table 1) was tested in a mixed-effects model with snakebite deaths per 100,000 people as the outcome variable. Directions differ for priors between the GBD 2019 and species-specific model if there was uncertainty if a prior direction would be true for all five venomous species models. An ensemble of the best-performing models was developed, which acted as the first-

stage linear prior in the ST-GPR model, weighted by out-of-sample RMSE. The model weights are defined by spatial distance across world regions and temporal distance.

We ran ST-GPR models for snakes, bees, scorpions, spiders, and a fifth other venom category to estimate the rate of death from all five animals for 204 countries, 23 age groups, males and females, for every year between 1980 and 2019, inclusive. To ensure the

ecological feasibility of our results, we zeroed out all locations that do not have endemic venomous snakes of medical importance, according to the WHO Snakebite Information and Data Platform[31]. Countries with zero snake deaths are given in Supplementary Table 4. The WHO Snakebite Information and Data Platform maps out the habitats of over 200 medically important venomous snakes, out of the 600 venomous snakes and 3000 overall species of snakes. The distribution map is based on published reference texts, scientific journals, museum collection databases, and consultations with zoologists and snakebite experts from around the world[31]. For each location, age, sex, and year demographic, we aggregated the results from all five different animals to derive the proportion of overall venomous animal deaths due to snakebites. This proportion was applied to the GBD 2019 venomous animal contact results from 1990 to 2019 to calculate the snakebite-specific mortality rate. Figure 5a displays the GBD 2019 all-ages rate of death from venomous animal contact, while Fig. 5b displays the proportion of those deaths due to just snakebite.

### Extrapolation, age-standardization, and forecasting for 2020 to 2050 estimates

Estimates for GBD 2019 span from 1990 to 2019. To estimate the number of deaths due to snakebite envenoming from 2020 to 2050 in 10-year intervals, we input the snakebite envenoming results into a regression with year and age as predictors. We conducted each regression by sex and region separately and added a cubic spline on age. Each sex- and region-specific regression was run 1000 times, and the resulting coefficients were used to predict rates in the years 2020, 2030, 2040, and 2050. Predicted rates were multiplied by the forecasted population and standardized using the GBD 2019 standard population[40]. No steps were made to align GBD 2019 mortality estimates with the predicted forecast from 2020, and predictions were made on the average annualized rate of change and the age-sex demographic composition of each region.

### Socio-demographic Index

SDI is a summary measure of development, taking into account a country's total fertility rate for women younger than 25 years, educational attainment, and lag-distributed income per capita. Methods to produce SDI are discussed elsewhere[15].

### GATHER compliance

This study complies with the Guidelines for Accurate and Transparent Health Estimates Reporting (GATHER) recommendations (Supplementary Information pp 20-22)[41].

### Reporting summary

Further information on research design is available in the Nature Research Reporting Summary linked to this article.

## Data availability

The findings from this study were produced using data available in public online repositories or in the published literature, data that are publicly available on request from the data provider, and data that are not publicly available due to restrictions by the data provider and which were used under license for the current study. Details on data sources can be found on the GHDx website, including information about the data provider and links to where the data can be accessed or requested (where available) at https://ghdx.healthdata.org/gbd-2019/data-input-sources?components=4&causes=710&locations=1. Citations for all 2657 input sources are available for download using the "download citations CSV" button on the linked page and can be viewed in Supplementary Data. Download source metadata (5,889,558 rows) are likewise available for download using the "download source metadata CSV" button. All available information for each input source is available by selecting the source from the alphabetically ordered list. Input sources can be filtered by location using the "locations" drop-down menu. Further information regarding the sources and how to obtain them is available upon request. We have also provided maps of the data included in our models in Supplementary Figs. 1–3. Source data are provided with this paper for Figs. 1–5. Information on whether or not venomous snakes inhabited a country was extracted from the World Health Organization Snakebite Information and Data Platform, which is available here: https://www.who.int/teams/control-of-neglected-tropical-diseases/snakebite-envenoming/snakebite-information-and-data-platform/overview#tab=tab_1. Source data are provided with this paper.

## Code availability

Our study follows the Guidelines for Accurate and Transparent Health Estimate Reporting (GATHER; Supplementary Table 7). All code used for the GBD 2019 analyses is publicly available online at https://ghdx.healthdata.org/gbd-2019/code and custom code for the snakebite envenomation analysis is publicly available online at https://github.com/nlr4002/Snakebite_Envenomation.

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

## Acknowledgements

This study was primarily funded by the Bill & Melinda Gates Foundation. M Ausloos and C Herteliu are partially supported by a grant from the Romanian National Authority for Scientific Research and Innovation, CNDS-UEFISCDI, project number PN-III-P4-ID-PCCF-2016-0084. F Carvalho and E Fernandes acknowledge Fundação para a Ciência e a Tecnologia, I.P., in the scope of the project UIDP/04378/2020 and UIDB/04378/2020 of the Research Unit on Applied Molecular Biosciences UCIBIO, and the project LA/P/0140/2020 of the Associate Laboratory Institute for Health and Bioeconomy i4HB; and FCT/MCTES (Ministério da Ciência, Tecnologia e Ensino Superior) through the project UIDB/50006/2020. K Krishan is supported by the UGC Centre of Advanced Study (Phase II), awarded to the Department of Anthropology, Panjab University, Chandigarh, India. I Landires is a member of the Sistema Nacional de Investigación (SNI), which is supported by Panama's Secretaría Nacional de Ciencia, Tecnología e Innovación (SENACYT). A M Samy acknowledges the support from the Egyptian Fulbright Mission Program. The funders of the study had no role in study design, data collection, data analysis, data interpretation, or writing of the final report. The corresponding author had full access to all the data in the study and had final responsibility for the decision to submit it for publication.

## Author contributions

N.L.S.R., E.B.H., M.N., S.L.J., D.M.P., S.I.H., T.V., and K.L.O. managed the estimation or publications process. N.L.S.R., R.D., L.D., S.L.J., D.M.P., S.I.H., T.V., and K.L.O. wrote the first draft of the manuscript. N.L.S.R., E.K.J., and S.M.Z. had primary responsibility for applying analytical methods to produce estimates. N.L.S.R., E.K.J., S.M.Z., and K.L.O. had primary responsibility for seeking, cataloging, extracting, or cleaning data and for designing or coding figures and tables. E.K.J., A.A., F.A., C.L.A., D.A., J.A., M.A., A.B., D.C., H.T.D., L.E., A.E., F.F., A.M.G., B.G.G., A.G., M.G., C.H., M.N.K., R.K., K.K., G.A.K., N.M., R.G.M., T.R.M., A.H.M., F.N.M., M.Moradi., C.T.N., H.L.T.N., J.R.P., H.Q.P., Z.Q.S., V.R., S.J.R., D.L.R., S.R., A.M.S., M.S., D.C.S., S.S., M.A.S., V.Y.S., A.A.S., A.S., M.A.S., M.R.T., B.X.T., R.S.T., M.N., R.D., L.D., D.M.P., C.J.L.M., and K.L.O. provided data or critical feedback on data sources. E.B.H., D.A., X.D., A.E., F.E., A.M.G., M.N.K., A.M., A.H.M., M.Moradi., Z.Q.S., R.C.R., A.M.S., M.N., D.M.P., C.J.L.M., T.V., and K.L.O. developed methods or computational machinery. F.A., V.A., R.A., C.L.A., D.A., J.A., M.A., A.F.A., A.D.B., A.Bhalla., N.B., P.B., S.B., A.Bijani., A.Bboloor., T.C., D.C., X.D., A.A.D., H.T.D., A.E., F.E., F.F., I.F., M.F., R.C.F., A.M.G., B.G.G., B.G., A.G., M.G., S.H., S.E.H., K.H., C.H., O.S.I., M.M.I., J.J., T.K., N.K., E.A.K., M.N.K., R.K., K.K., G.A.K., N.K., S.S.L., V.M., N.M., L.B.M., R.G.M., T.J.M., T.R.M., A.M., A.H.M., F.N.M., M.Moradi., C.T.N., H.L.T.N., S.M.O., J.R.P., H.Q.P., Z.Q.S., N.R., A.R., V.R., S.J.R., P.R., D.L.R., S.R., A.M.S., M.S., D.C.S., M.A.S., V.Y.S., A.A.S., A.S., M.A.S., R.T., M.R.T., B.X.T., R.S.T., D.Z.V., Z.Z., M.N., R.D., L.D., D.M.P., C.J.L.M., S.I.H., T.V., and K.L.O. provided critical feedback on methods or results. F.A., R.A., C.L.A., J.A., M.A., S.M.B., S.B., F.C., D.C., R.A.S.C., H.T.D., A.E., F.E., E.F., I.F., M.F., A.M.G., B.G., A.G.,

C.H., O.S.I., J.J., E.A.K., M.N.K., K.K., I.L., M.Madadin., V.M., N.M., L.B.M., R.G.M., T.J.M., T.R.M., A.M., A.H.M., F.N.M., M.Moradi., V.C.N., C.T.N., H.L.T.N., V.N., J.R.P., H.Q.P., M.Pinheiro., M.Pirestani., Z.Q.S., N.R., A.R., V.R., S.J.R., D.L.R., S.R., A.S., A.M.S., D.C.S., V.Y.S., A.A.S., M.A.S., M.R.T., B.X.T., D.Z.V., M.N., D.M.P., S.I.H., T.V., and K.L.O. drafted the work or revised it critically for important intellectual content. A.H.M., M.N., L.D., C.J.L.M., S.I.H., and T.V. managed the overall research enterprise.

## Competing interests
The authors declare no competing interests.

## Additional information

## GBD 2019 Snakebite Envenomation Collaborators

Nicholas L. S. Roberts [1], Emily K. Johnson [2], Scott M. Zeng[2], Erin B. Hamilton[2], Amir Abdoli [3], Fares Alahdab [4], Vahid Alipour[5,6], Robert Ancuceanu[7], Catalina Liliana Andrei[8], Davood Anvari[9,10], Jalal Arabloo[5], Marcel Ausloos[11,12], Atalel Fentahun Awedew [13], Ashish D. Badiye [14], Shankar M. Bakkannavar [15], Ashish Bhalla[16], Nikha Bhardwaj [17], Pankaj Bhardwaj [18,19], Soumyadeep Bhaumik [20,21], Ali Bijani [22], Archith Boloor [23], Tianji Cai[24], Felix Carvalho [25], Dinh-Toi Chu [26], Rosa A. S. Couto[27], Xiaochen Dai[2,28], Abebaw Alemayehu Desta[29], Hoa Thi Do[30], Lucas Earl[2], Aziz Eftekhari[31,32], Firooz Esmaeilzadeh[33], Farshad Farzadfar[34], Eduarda Fernandes[35], Irina Filip[36,37], Masoud Foroutan [38,39], Richard Charles Franklin [40], Abhay Motiramji Gaidhane[41], Birhan Gebresillassie Gebregiorgis[42], Berhe Gebremichael[43], Ahmad Ghashghaee[5,44], Mahaveer Golechha[45], Samer Hamidi[46], Syed Emdadul Haque [47], Khezar Hayat[48,49], Claudiu Herteliu[12,50], Olayinka Stephen Ilesanmi[51,52], M. Mofizul Islam [53], Jagnoor Jagnoor[54], Tanuj Kanchan[55], Neeti Kapoor [14], Ejaz Ahmad Khan [56], Mahalaqua Nazli Khatib [57], Roba Khundkar[58], Kewal Krishan [59], G. Anil Kumar[60], Nithin Kumar[61], Iván Landires[62,63], Stephen S. Lim[2,28], Mohammed Madadin[64], Venkatesh Maled [65,66], Navid Manafi [67], Laurie B. Marczak[2], Ritesh G. Menezes [68], Tuomo J. Meretoja[69,70], Ted R. Miller [71,72], Abdollah Mohammadian-Hafshejani[73], Ali H. Mokdad[2,28], Francis N. P. Monteiro[74], Maryam Moradi [75], Vinod C. Nayak[15], Cuong Tat Nguyen[76], Huong Lan Thi Nguyen[76], Virginia Nuñez-Samudio[77,78], Samuel M. Ostroff[2,79], Jagadish Rao Padubidri[80], Hai Quang Pham[81], Marina Pinheiro[82], Majid Pirestani[83], Zahiruddin Quazi Syed [41], Navid Rabiee [84], Amir Radfar[85], Vafa Rahimi-Movaghar[86], Sowmya J. Rao[87], Prateek Rastogi [88], David Laith Rawaf[89,90], Salman Rawaf [91,92], Robert C. Reiner Jr. [2,28], Amirhossein Sahebkar[93,94], Abdallah M. Samy [95], Monika Sawhney[96], David C. Schwebel [97], Subramanian Senthilkumaran [98], Masood Ali Shaikh [99], Valentin Yurievich Skryabin [100], Anna Aleksandrovna Skryabina [101], Amin Soheili[102], Mark A. Stokes [103], Rekha Thapar[61], Marcos Roberto Tovani-Palone [104,105], Bach Xuan Tran[106], Ravensara S. Travillian[2], Diana Zuleika Velazquez[107], Zhi-Jiang Zhang [108], Mohsen Naghavi[2,28], Rakhi Dandona[2,28,60], Lalit Dandona[2,60,109], Spencer L. James[110], David M. Pigott [2,28], Christopher J. L. Murray[2,28], Simon I. Hay [2,28], Theo Vos [2,28] & Kanyin Liane Ong[2]

[1]Department of Medicine, Center for Global Health, Weill Cornell Medicine, New York, NY, USA. [2]Institute for Health Metrics and Evaluation, University of Washington, Seattle, WA, USA. [3]Zoonoses Research Center, Jahrom University of Medical Sciences, Jahrom, Iran. [4]Mayo Evidence-based Practice Center, Mayo Clinic Foundation for Medical Education and Research, Rochester, MN, USA. [5]Health Management and Economics Research Center, Iran University of Medical Sciences, Tehran, Iran. [6]Department of Health Economics, Iran University of Medical Sciences, Tehran, Iran. [7]Pharmacy Department, Carol Davila University of Medicine and Pharmacy, Bucharest, Romania. [8]Cardiology Department, Carol Davila University of Medicine and Pharmacy, Bucharest, Romania. [9]Department of Parasitology, Mazandaran University of Medical Sciences, Sari, Iran. [10]Department of Parasitology, Iranshahr University of Medical Sciences, Iranshahr, Iran. [11]School of Business, University of Leicester, Leicester, UK. [12]Department of Statistics and Econometrics, Bucharest University of Economic Studies, Bucharest, Romania. [13]Department of Surgery, Addis Ababa University, Addis Ababa, Ethiopia. [14]Department of Forensic Science, Government Institute of Forensic Science, Nagpur, India. [15]Department of Forensic Medicine and Toxicology, Manipal Academy of Higher Education, Manipal, India. [16]Department of Internal Medicine, Post Graduate Institute of Medical Education and Research, Chandigarh, India. [17]Department of Anatomy, Government Medical College Pali, Pali, India. [18]Department of Community Medicine and Family Medicine, All India Institute of Medical Sciences, Jodhpur, India. [19]School of

Public Health, All India Institute of Medical Sciences, Jodhpur, India. [20]Injury Division, The George Institute for Global Health, New Delhi, India. [21]The George Institute for Global Health, University of New South Wales, Sydney, NSW, Australia. [22]Social Determinants of Health Research Center, Babol University of Medical Sciences, Babol, Iran. [23]Department of Internal Medicine, Manipal Academy of Higher Education, Mangalore, India. [24]Department of Sociology, University of Macau, Macau, China. [25]Research Unit on Applied Molecular Biosciences (UCIBIO), University of Porto, Porto, Portugal. [26]Center for Biomedicine and Community Health, VNU-International School, Hanoi, Vietnam. [27]Department of Chemical Sciences, University of Porto, Porto, Portugal. [28]Department of Health Metrics Sciences, School of Medicine, University of Washington, Seattle, WA, USA. [29]Department of Surgical Nursing, University of Gondar, Gondar, Ethiopia. [30]Institute of Health Economics and Technology, Hanoi, Vietnam. [31]Russian Institute for Advanced Study, Moscow State Pedagogical University, Moscow, Russia. [32]Department of Surface Engineering, The John Paul II Catholic University of Lublin, Lublin, Poland. [33]Department of Public Health, Maragheh University of Medical Sciences, Maragheh, Iran. [34]Non-communicable Diseases Research Center, Tehran University of Medical Sciences, Tehran, Iran. [35]Associated Laboratory for Green Chemistry (LAQV), University of Porto, Porto, Portugal. [36]Psychiatry Department, Kaiser Permanente, Fontana, CA, USA. [37]School of Health Sciences, A.T. Still University, Mesa, AZ, USA. [38]Department of Medical Parasitology, Abadan University of Medical Sciences, Abadan, Iran. [39]Faculty of Medicine, Abadan University of Medical Sciences, Abadan, Iran. [40]School of Public Health, Medical, and Veterinary Sciences, James Cook University, Douglas, QLD, Australia. [41]Department of Community Medicine, Datta Meghe Institute of Medical Sciences, Wardha, India. [42]Department of Nursing, Debre Berhan University, Debre Berhan, Ethiopia. [43]School of Public Health, Haramaya University, Harar, Ethiopia. [44]Student Research Committee, Iran University of Medical Sciences, Tehran, Iran. [45]Health Systems and Policy Research, Indian Institute of Public Health, Gandhinagar, India. [46]School of Health and Environmental Studies, Hamdan Bin Mohammed Smart University, Dubai, United Arab Emirates. [47]Institute of Statistical Research and Training, University of Dhaka, Dhaka, Bangladesh. [48]Institute of Pharmaceutical Sciences, University of Veterinary and Animal Sciences, Lahore, Pakistan. [49]Department of Pharmacy Administration and Clinical Pharmacy, Xian Jiaotong University, Xian, China. [50]School of Business, London South Bank University, London, UK. [51]Department of Community Medicine, University of Ibadan, Ibadan, Nigeria. [52]Department of Community Medicine, University College Hospital, Ibadan, Ibadan, Nigeria. [53]School of Psychology and Public Health, La Trobe University, Melbourne, VIC, Australia. [54]The George Institute for Global Health, University of New South Wales, New Delhi, India. [55]Department of Forensic Medicine and Toxicology, All India Institute of Medical Sciences, Jodhpur, India. [56]Department of Epidemiology and Biostatistics, Health Services Academy, Islamabad, Pakistan. [57]Global Evidence Synthesis Initiative, Datta Meghe Institute of Medical Sciences, Wardha, India. [58]Nuffield Department of Surgical Sciences, University of Oxford, Oxford, UK. [59]Department of Anthropology, Panjab University, Chandigarh, India. [60]Public Health Foundation of India, Gurugram, India. [61]Department of Community Medicine, Manipal Academy of Higher Education, Mangalore, India. [62]Unit of Genetics and Public Health, Institute of Medical Sciences, Las Tablas, Panama. [63]Ministry of Health, Herrera, Panama. [64]Pathology Department, Imam Abdulrahman Bin Faisal University, Dammam, Saudi Arabia. [65]Department of Forensic Medicine, Shri Dharmasthala Manjunatheshwara University, Dharwad, India. [66]Department of Forensic Medicine, Rajiv Gandhi University of Health Sciences, Bangalore, India. [67]Doheny Eye Institute, University of California Los Angeles, Los Angeles, CA, USA. [68]Forensic Medicine Division, Imam Abdulrahman Bin Faisal University, Dammam, Saudi Arabia. [69]Breast Surgery Unit, Helsinki University Hospital, Helsinki, Finland. [70]University of Helsinki, Helsinki, Finland. [71]Pacific Institute for Research & Evaluation, Calverton, MD, USA. [72]School of Public Health, Curtin University, Perth, WA, Australia. [73]Department of Epidemiology and Biostatistics, Shahrekord University of Medical Sciences, Shahrekord, Iran. [74]Department of Forensic Medicine & Toxicology, A.J. Institute of Medical Sciences and Research Centre, Mangalore, India. [75]Iran University of Medical Sciences, Iran University of Medical Sciences, Tehran, Iran. [76]Institute for Global Health Innovations, Duy Tan University, Hanoi, Vietnam. [77]Unit of Microbiology and Public Health, Institute of Medical Sciences, Las Tablas, Panama. [78]Department of Public Health, Ministry of Health, Herrera, Panama. [79]Henry M Jackson School of International Studies, University of Washington, Seattle, Washington, USA. [80]Kasturba Medical College, Mangalore, Manipal Academy of Higher Education, Manipal, India. [81]Center of Excellence in Behavioral Medicine, Nguyen Tat Thanh University, Ho Chi Minh City, Vietnam. [82]Department of Chemistry, University of Porto, Porto, Portugal. [83]Department of Parasitology and Entomology, Tarbiat Modares University, Tehran, Iran. [84]Department of Physics, Sharif University of Technology, Tehran, Iran. [85]College of Medicine, University of Central Florida, Orlando, FL, USA. [86]Sina Trauma and Surgery Research Center, Tehran University of Medical Sciences, Tehran, Iran. [87]Department of Oral Pathology, Srinivas Institute of Dental Sciences, Mangalore, India. [88]Department of Forensic Medicine and Toxicology, Manipal Academy of Higher Education, Mangalore, India. [89]WHO Collaborating Centre for Public Health Education and Training, Imperial College London, London, UK. [90]University College London Hospitals, London, UK. [91]Department of Primary Care and Public Health, Imperial College London, London, UK. [92]Academic Public Health England, Public Health England, London, UK. [93]Applied Biomedical Research Center, Mashhad University of Medical Sciences, Mashhad, Iran. [94]Biotechnology Research Center, Mashhad University of Medical Sciences, Mashhad, Iran. [95]Department of Entomology, Ain Shams University, Cairo, Egypt. [96]Department of Public Health Sciences, University of North Carolina at Charlotte, Charlotte, NC, USA. [97]Department of Psychology, University of Alabama at Birmingham, Birmingham, AL, USA. [98]Emergency Department, Manian Medical Centre, Erode, India. [99]Independent Consultant, Karachi, Pakistan. [100]Department No.16, Moscow Research and Practical Centre on Addictions, Moscow, Russia. [101]Therapeutic Department, Balashiha Central Hospital, Balashikha, Russia. [102]Nursing Care Research Center, Semnan University of Medical Sciences, Semnan, Iran. [103]Department of Psychology, Deakin University, Burwood, VIC, Australia. [104]Department of Pathology and Legal Medicine, University of São Paulo, Ribeirão Preto, Brazil. [105]Modestum LTD, London, UK. [106]Department of Health Economics, Hanoi Medical University, Hanoi, Vietnam. [107]Faculty of Veterinary Medicine and Zootechnics, Autonomous University of Sinaloa, Culiacán Rosales, Mexico. [108]School of Medicine, Wuhan University, Wuhan, China. [109]Indian Council of Medical Research, New Delhi, India. [110]Personalized Healthcare, Genentech, San Francisco, CA, USA. ✉e-mail: nlr4002@med.cornell.edu

