## [Peer Review File · Nature Communications]

Reviewers' Comments:

Reviewer #1:

Remarks to the Author:

Dear authors,

Thank you very much for the opportunity to review this very interesting and useful paper.

Major:

1. One general comment : I am surprised that the N deaths is reduced by half compared to the estimations by Gutierrez et al. (Nature Dis Prim) and by WHO (website 2021) : 81,000-138,000 deaths. Therefore I am worried that you under-estimate the burden with the GBD methodology, maybe prioritizing hospital deaths, and purely VE and VA, vs out-of-hospital deaths (community based studies and random cluster surveys...).

Experts report that deaths in India alone are 58,000/year btw 2010 and 2019(Suraweera et al. 2020 <https://elifesciences.org/articles/54076>) and is relatively stable around 61,200 since 2014 due to better hospital data reporting and better access to hospitals (vs community deaths after access to traditional healers).

Of course we understand that you wish to respect thoroughly the GBD of disease methodology.

2. The results chapter is very much centered on India, with very little detail on the large burden found in other South/South-East Asian countries, East Africa (South Sudan and Ethiopia), Central and West Africa (Nigeria, Benin, Ghana, Cameroon), the MENA region (e.g Yemen and Iran).

3. Lines 195-203: you easily recognize that community-based deaths and non-recorded deaths in remote rural centers probably cause an important under-estimation. We suggest that an adjusted mortality estimated is attempted: in India, Nepal , Sri Lanka, Cameroon, recent nationwide cluster random surveys show that deaths occurring in the community or at home are more than double that recorded by hospital data. This suggests that your report is an underestimation.

See Preprint with the Lancet on Nepal (cluster random sampling from 63,000 participants): estimated deaths 20/100,000/year representing 37,661 cases and 2949 deaths/yr in Nepal's Terai. (compared to 20,000 cases and 1000 deaths/yr according to MOH).

https://papers.ssrn.com/sol3/papers.cfm?abstract_id=3867686

A similar study (same methodology, still unpublished) on 61,000 participants in Cameroon also showed a very high incidence and a mortality rate of 32/100,000/yr. (communication #71, oral presentation SFMTSI/SAV/SAOP congress 1-3Dec2021, available PDF on request). Other subnational studies suggest similar mortality rates in neighbouring countries. Therefore, perhaps, a multiplication ratio could be applied to the hospital-based routine data in African countries.

An excellent population-based study in Sri Lanka shows "The crude overall community incidence of snakebite, envenoming and mortality were 398 (95% CI: 356-441), 151 (130-173) and 2.3 (0.2-4.4) per 100000 population, respectively."

<https://www.ncbi.nlm.nih.gov/pmc/articles/PMC4938527/>

-> This is higher than the ASMR of 0.22 that you report in Sri Lanka based on a 2006 study. More incidence studies have been published in Sri Lanka.

These estimates of the global estimates and global risk are useful to address the more globale impact - beyond your data.

*Pintor et al 2021 Toxicon <https://www.ncbi.nlm.nih.gov/pmc/articles/PMC8350508/>

Longbottom et al .

Lancet <https://www.ncbi.nlm.nih.gov/pmc/articles/PMC8350508/#bib151>

Minor comments:

1. Please replace "Venomous snakebite" by "Snakebite Envenoming" (SBE) the standard name of this NTD listed by the WHO NTD department and by the WHA resolution.

2. Conflict areas (Yemen, South-Sudan, CAR, parts of Nigeria and Ethiopia) are not mentioned as high-risk of snakebite due to displacement and outdoor exposure and high-risk of under-reporting. See Alcoba et al. (Toxicon 2021).

3. Increase the N of references from countries outside India to represent the "global burden", and the global challenges in the discussion (maybe too many examples from the challenges in India could be interpreted as a bias).

Wishing you the very best for this submission !

Reviewer #2:

Remarks to the Author:

This study uses a global data repository and modeling tools to quantify mortality and years of life lost due to venomous snakebites in 204 countries and territories over a span of nearly 30 years. The findings are consistent with previously published data.

As discussed by the authors, these types of analyses are limited in their accuracy due to the many assumptions in calculations and a marked paucity of reliable data on the number of snakebites and associated deaths from snakebite endemic countries. Consequently, true regional and global figures may differ considerably from the computed estimates. This is especially true for regions like sub-Saharan Africa which lack robust nationwide community studies. Similarly, the authors estimate 54,600 deaths due to venomous snakebite in south Asia. However, a recent nationally representative mortality study from India estimated 58,000 annual deaths from India alone. Large, well-designed, representative community-based epidemiological studies are needed to better assess the magnitude of venomous snakebite and associated deaths in endemic regions.

REVIEWER COMMENTS

Reviewer #1 (Remarks to the Author):

Dear authors,

Thank you very much for the opportunity to review this very interesting and useful paper.

Major:

1. One general comment : I am surprised that the N deaths is reduced by half compared to the estimations by Gutierrez et al. (Nature Dis Prim) and by WHO (website 2021) : 81,000-138,000 deaths. Therefore I am worried that you under-estimate the burden with the GBD methodology, maybe prioritizing hospital deaths, and purely VE and VA, vs out-of-hospital deaths (community based studies and random cluster surveys...).

We thank the reviewer for their comment and agree it is important to demonstrate why our results are lower than commonly cited global estimates of snakebite deaths.

The two articles cited by the WHO and Gutierrez et al. giving estimates of 81,000 and 138,000 deaths are from meta-analyses by Chippaux in 1998¹ and Kasturiratne in 2008.² While these two articles were foundational in describing the global burden of snakebites, they required multiple assumptions and extensive extrapolation from sparse epidemiological data. After investigating these two articles further, we believe our results are less of an underestimate than it appears and are within the same range.

Chippaux calculated his estimate of global deaths from aggregating the available household surveys, hospital records, and health authority statistics in 1998. While his assessment was fundamental to the understanding of snakebite envenomation as a serious global health issue, his methodology to estimate deaths was crude and based primarily on his own expert opinion instead of a robust epidemiological method. For example, in Asia he claims based on his literature review “there may be up to 4 million snake-bites each year, of which almost 50% are envenomed. Approximately half of the victims reach hospital and the annual number of deaths resulting can be estimated at 100,000.” In Africa, he says, “probably 1 million snake-bites occur every year involving 500,000 envenomations, of which 40% are hospitalized. It is likely that about 20,000 deaths per year occur as a result, although less than 10,000 are reported by health services.” After a literature review and estimation of every region, he reached a total of 125,345 deaths. His analysis relied extensively on his own approximations on the regional incidence and mortality of snakebites. With improvements over the last 25 years in data collection and epidemiological modeling methods, it is possible to estimate the global number of deaths from snakebite envenomation more accurately and more based on data than expert opinion.

Kasturiratne et al. used a similar method as Chippaux, extracting data from 68 countries to estimate the incidence and case-fatality rate of snakebite envenomation. Like any global estimation process, their methods had assumptions, such as using the mortality rate from neighboring countries within a region that had data as the estimate in countries without data, to estimate an incidence and mortality value for every country. They also did not include time into

their estimate and used the most recent data after 1985 that they could find as the current rate of death for a country (up to 2008). This method would not account for any progress towards preventing and treating snakebites and could have overestimated deaths. Alternatively, it could also underestimate deaths in locations that have had significant ecological changes that have increased human-wildlife conflict. By modeling snakebite deaths over time and location, we rely less strongly on assumptions while utilizing variables such as education, income, and climate to guide our estimates instead of considering them unchanged over time or equivalent across borders.

Importantly, Kasturiratne et al. published both a low and high estimate in their article. Their low estimate for the global number of deaths was 20,000 and their high estimate was 94,000 deaths. Our global estimate falls above the mean of their two estimates (63,400 vs. 57,000), and we do not believe it should be considered an underestimation if both resulting estimates of Kasturiratne et al.'s analysis were reported in the literature. However, their "high" estimate for deaths is more commonly cited.

1: Chippaux JP. Snake-bites: appraisal of the global situation. *Bull World Health Organ* 1998; **76**: 515–24.

2: Kasturiratne A, Wickremasinghe AR, de Silva N, et al. The Global Burden of Snakebite: A Literature Analysis and Modelling Based on Regional Estimates of Envenoming and Deaths. *PLoS Med* 2008; **5**: e218.

Experts report that deaths in India alone are 58,000/year btw 2010 and 2019 (Suraweera et al. 2020 <https://elifesciences.org/articles/54076>) and is relatively stable around 61,200 since 2014 due to better hospital data reporting and better access to hospitals (vs community deaths after access to traditional healers).

Of course we understand that you wish to respect thoroughly the GBD of disease methodology.

In our analysis, we use the exact same verbal autopsy data that Suraweera et al. used, along with additional verbal autopsy data from the country. Our results are slightly less due to various reasons. First, Suraweera et al. estimated that snakebite mortality had a steady decrease from 2001 to 2011, an increase from 2011 to 2014, and then they estimated that the disease burden from 2015 to 2019 decreased, according to the caption from Table 2. Our modeling process attempted to smooth out this trend over time based on the point estimates and underlying uncertainty in these surveys. Our process also incorporated variables such as education, healthcare access and quality, and social development of the country, which have improved in India and have a negative correlation with snakebite mortality. This led to our estimate of a more constant decrease over time.

Suraweera et al. also estimated an age-standardized death rate of 4.2 per 100,000 in 2011, and a subsequent increase of 15% in 2014 to 4.8 per 100,000. We do not consider our estimate of 4.0 deaths per 100,000 (95% uncertainty interval from 2.31 to 5.01) significantly different than their results. Their age-standardization weights are also based on the Indian census reference weights,

while ours are based on GBD population reference weights, which more closely resemble the WHO global standard age weights. This difference in age weighting could also account for some of the difference between our age-standardized mortality rate and theirs.

2. The results chapter is very much centered on India, with very little detail on the large burden found in other South/South-East Asian countries, East Africa (South Sudan and Ethiopia), Central and West Africa (Nigeria, Benin, Ghana, Cameroon), the MENA region (e.g Yemen and Iran).

We thank the reviewers for raising this concern, and agree that the results from African countries should be discussed more directly in the manuscript. We have added a paragraph in the results on the burden of snakebite envenoming in sub-Saharan Africa (lines 92 to 99). We have also expanded on our discussion section to include more research from sub-Saharan Africa (lines 132-148).

3. Lines 195-203: you easily recognize that community-based deaths and non-recorded deaths in remote rural centers probably cause an important under-estimation. We suggest that an adjusted mortality estimated is attempted: in India, Nepal , Sri Lanka, Cameroon, recent nationwide cluster random surveys show that deaths occurring in the community or at home are more than double that recorded by hospital data. This suggests that your report is an underestimation. See Preprint with the Lancet on Nepal (cluster random sampling from 63,000 participants): estimated deaths 20/100,000/year representing 37,661 cases and 2949 deaths/yr in Nepal's Terai. (compared to 20,000 cases and 1000 deaths/yr according to MOH).

https://papers.ssrn.com/sol3/papers.cfm?abstract_id=3867686

A similar study (same methodology, still unpublished) on 61,000 participants in Cameroon also showed a very high incidence and a mortality rate of 32/100,000/yr. (communication #71, oral presentation SFMTSI/SAV/SAOP congress 1-3Dec2021, available PDF on request). Other subnational studies suggest similar mortality rates in neighbouring countries. Therefore, perhaps, a multiplication ratio could be applied to the hospital-based routine data in African countries.

An excellent population-based study in Sri Lanka shows "The crude overall community incidence of snakebite, envenoming and mortality were 398 (95% CI: 356–441), 151 (130–173) and 2.3 (0.2–4.4) per 100000 population, respectively." <https://www.ncbi.nlm.nih.gov/pmc/articles/PMC4938527/>

-> This is higher than the ASMR of 0.22 that you report in Sri Lanka based on a 2006 study. More incidence studies have been published in Sri Lanka.

We agree with the reviewer that relying on official government health statistics and hospital data could lead to an underestimation in our data. In Sri Lanka, we relied on ICD-coded nationally representative and complete vital registration data from 2006, 2009, and 2010. It is possible these

inputs are underestimating the number of deaths in Sri Lanka due to our reliance on official health statistics.

The study by Ediriweera *et al.* shows that verbal autopsy surveys also have limitations in their ability to estimate venomous snakebite mortality with precision, due to the relative scarcity of snakebite envenoming deaths. The study's estimates of incidence and envenoming have relatively precise confidence intervals, with estimates of 398 (356-441) and 151 (130-173), respectively. However, their estimate of mortality has very wide confidence intervals, with an estimate of 2.3 (0.2-4.4) deaths per 100,000, or an estimate of 45 to 884 deaths. In the study, they only measured 5 deaths out of 165,665 people in 12 months. In the conclusion paragraph of their abstract and discussion, they only mention incidence and do not mention their mortality estimates, likely because of their more robust findings of incidence compared to mortality. Ediriweera *et al.*'s survey is very high-quality and demonstrates that estimating the true magnitude of mortality is difficult compared to measuring incidence of snakebites. Our estimate is within the confidence interval of the estimate by Ediriweera, and we have addressed the differences directly in our manuscript (lines 201 to 216), because we believe their analysis shows the difficulty of attaining a true estimate of mortality despite a high-quality community-based survey.

We read the recent article on snakebite envenomation in the Nepali Terai with great interest. Their estimate of 22.4 (95% confidence interval 11.9 to 42.1) deaths per 100,000 were some of the greatest mortality estimates we had ever seen. According to our estimates from GBD 2019, that would place snakebite envenomation as the 9th leading cause of death in Nepal, barely behind TB (24.6 deaths per 100,000) and asthma (22.7 deaths per 100,000), and ahead of diarrheal diseases (18.5 deaths per 100,000) and chronic kidney disease (16.8 deaths per 100,000). It would make venomous snakebite the greatest cause of death among injuries, ahead of falls (12.7 deaths per 100,000), self-harm (11.6 deaths per 100,000), and road injuries (8.9 deaths per 100,000).

The results cited above from Cameroon are also very high compared to other published literature, especially in comparison with the mortality rate of other diseases in the country. According to the GBD, a mortality rate of 32 per 100,000 would make venomous snakebite the 8th leading cause of death in Cameroon, behind ischemic heart disease (36.5 deaths per 100,000) and ahead of TB (22.5 per 100,000), road injuries (22.0 per 100,000), and diabetes (17.8 per 100,000).

The GBD is an iterative process, and we are hoping to include snakebite envenoming as one of the causes of death that are included in our annual releases. Going forward, we will investigate these two articles and hopefully include them and other community surveys as data inputs into the GBD to improve the accuracy of our estimates. However, the results from these two studies are substantially greater than any previously published estimate of venomous snakebite to our knowledge. To include them we would need to investigate their methodology and representativeness further to better understand the degree of comparability between vital registration, hospital data sources, and community-based surveys. While venomous snakebite is an underappreciated problem in South Asia and sub-Saharan Africa, we would need to conduct

further analyses to validate that they are a bigger problem than major known causes of death like TB, self-harm, and road injuries.

We have expanded our discussion to include a reference to this Nepali survey as an example that newer verbal autopsy surveys are showing substantially higher mortality rates from snakebite envenoming than previously published in the literature (lines 220 to 231). Using these surveys, hopefully we will have a more robust dataset to compare vital registration and hospital data with household surveys, and be able to have improved estimates in regions like sub-Saharan Africa and South Asia.

These estimates of the global estimates and global risk are useful to address the more global impact - beyond your data.

*Pintor et al 2021 Toxicon <https://www.ncbi.nlm.nih.gov/pmc/articles/PMC8350508/>

Thank you for bringing this article to our attention. It does a wonderful job summarizing the need for advanced and regular data collection as well as more robust geospatial analyses to better capture the true burden of disease of snakebite envenoming and guide antivenom production and distribution. We have included reference to it in our introduction as well as discussion, because we believe it describes the need for an analysis like ours using advanced statistical modeling and updated global data sources to estimate the global burden of snakebite envenomation (lines 44-46). Through the use of the GBD data repository and modeling infrastructure, we hope to demonstrate that venomous snakebite can be analyzed with the same robustness as other major causes of disease.

Longbottom et al. Lancet <https://www.ncbi.nlm.nih.gov/pmc/articles/PMC8350508/#bib151>

We agree with the reviewer that this study was an important step in using granular geospatial analyses at a global level, and provides important insights into what populations are most vulnerable. We have a whole paragraph devoted to comparing our results with Longbottom et al.'s work because we found the two highly complementary. We found high rates of death in many areas of the world that Longbottom described as vulnerable to venomous snakebite. We also found high rates of death in many areas of the world that Longbottom did not find were vulnerable, especially in India. This is because the Big 4 venomous snakes in India have antivenoms that are effective, and India has many large population centers, which were two factors that Longbottom et al. considered to decrease the vulnerability of the population to snakebite envenoming. However, our results demonstrate that snakebite envenoming is a major problem in India despite these factors, and that further interventions are required to make antivenom more accessible to the population at risk.

Minor comments:

1. Please replace "Venomous snakebite" by "Snakebite Envenoming" (SBE) the standard name of this NTD listed by the WHO NTD department and by the WHA resolution.

We appreciate the reviewer for pointing this out and ensuring our terminology is aligned with the WHO's. We have corrected the term venomous snakebite throughout the manuscript.

2. Conflict areas (Yemen, South-Sudan, CAR, parts of Nigeria and Ethiopia) are not mentioned as high-risk of snakebite due to displacement and outdoor exposure and high-risk of under-reporting. See Alcoba et al. (Toxicon 2021).

We thank the reviewer for bringing up this important point. The association between snakebite and conflict is an important factor to consider, and we have included it in our additional discussion paragraph on burden of snakebite envenoming in sub-Saharan Africa (lines 132 to 148).

3. Increase the N of references from countries outside India to represent the "global burden", and the global challenges in the discussion (maybe too many examples from the challenges in India could be interpreted as a bias).

We agree with the reviewer that our article is meant to describe the global burden and was too overly focused on India. We hope that our additional points about the burden of snakebite envenoming in sub-Saharan Africa, Sri Lanka, and Nepal help make our manuscript appear less focused on India alone. We kept in the part on the Big Four snakes in India, because we think it is an important illustration of the need for health system infrastructure improvements to increase access to antivenom in rural areas, as well as explain the difference between our high Indian mortality results and Longbottom's low Indian vulnerability results, to give more context on the factors that increase the risk of snakebite envenoming mortality.

Wishing you the very best for this submission !

Thank you! We deeply appreciate your time and consideration to help improve our manuscript.

Reviewer #2 (Remarks to the Author):

This study uses a global data repository and modeling tools to quantify mortality and years of life lost due to venomous snakebites in 204 countries and territories over a span of nearly 30 years. The findings are consistent with previously published data.

As discussed by the authors, these types of analyses are limited in their accuracy due to the many assumptions in calculations and a marked paucity of reliable data on the number of snakebites and associated deaths from snakebite endemic countries. Consequently, true regional and global figures may differ considerably from the computed estimates. This is especially true for regions like sub-Saharan Africa which lack robust nationwide community studies. Similarly, the authors estimate 54,600 deaths due to venomous snakebite in south Asia. However, a recent nationally representative mortality study from India estimated 58,000 annual deaths from India alone.

Large, well-designed, representative community-based epidemiological studies are needed to better assess the magnitude of venomous snakebite and associated deaths in endemic regions.

We thank the reviewer for their comments. We agree that large, well-designed, representative community-based epidemiological studies are needed to better understand the burden due to venomous snakebites. While like any modeling process, our methodology has assumptions, it has been validated extensively and has been built to create the most accurate estimates in countries that lack their own robust data by leveraging the rest of the global dataset and different variables such as income, education, climate, and others.

With regards to the results from Suraweera et al. in India, our analysis used the exact same verbal autopsy data as their study, along with additional verbal autopsy data from the country. Our results are slightly less due to various reasons. First, Suraweera et al. estimated that snakebite mortality had a steady decrease from 2001 to 2011, a significant increase from 2011 to 2014, and then estimated that the disease burden decreased from 2015 to 2019, according to the caption from Table 2 in their article. Our modeling process attempted to smooth out this trend over time based on the point estimates and underlying uncertainty in the surveys. Our modeling process also incorporated variables such as education, healthcare access and quality, and social development of the country, which have improved in India and have a negative correlation with snakebite mortality. This led to our estimate of a more constant decrease over time.

We fully agree that robust community-based epidemiological studies are required to fully understand the burden of snakebite envenomation. The GBD is an iterative process, and we plan to include snakebite envenomation in our annual release in order to continually update our estimates as new studies come out and demonstrate where gaps in epidemiological surveillance of snakebite persist.

Reviewers' Comments:

Reviewer #1:

Remarks to the Author:

Dear authors and editors,

I has been a true pleasure to read your detailed answers to my multiple questions and challenges on the methodology, scarcity/heterogeneity of data, and comparability of findings between hospital-data and community random surveys.

I am impressed by the level of attention given to specific points and by your improvements to include more studies and more data from Africa and Asia to make it less focused on India.

I understand the need to strictly follow the GBD methodology, and cannot challenge this one, which has been proved effective and comparable across diseases and health systems.

I appreciate that you recognized some limitations of your approach, and that we will need to have better understanding of the out-of-hospital incidence and mortality (indeed some have wider 95%CI) but often being more than 2-3 times higher than facility-based mortality. However my observations is limited few truly representative studies (5-6 nationwide studies) which is probably too few for robust extrapolations to global estimates using the GBD system or any other.

To conclude, such studies are essential to show the massive burden of snakebite envenoming (SBE), and it was extremely carefully performed, hopefully the next one will include more precise data from areas which are non/under-reporting. I do not have any further reasons to delay the publication of this excellent work.

Many thanks to the editors for the opportunity to review and read this paper before others!

Reviewer #2:

Remarks to the Author:

No further comments

REVIEWERS' COMMENTS

Reviewer #1 (Remarks to the Author):

Dear authors and editors,

I has been a true pleasure to read your detailed answers to my multiple questions and challenges on the methodology, scarcity/heterogeneity of data, and comparability of findings between hospital-data and community random surveys.

I am impressed by the level of attention given to specific points and by your improvements to include more studies and more data from Africa and Asia to make it less focused on India.

I understand the need to strictly follow the GBD methodology, and cannot challenge this one, which has been proved effective and comparable across diseases and health systems.

I appreciate that you recognized some limitations of your approach, and that we will need to have better understanding of the out-of-hospital incidence and mortality (indeed some have wider 95%CI) but often being more than 2-3 times higher than facility-based mortality. However my observations is limited few truly representative studies (5-6 nationwide studies) which is probably too few for robust extrapolations to global estimates using the GBD system or any other.

To conclude, such studies are essential to show the massive burden of snakebite envenoming (SBE), and it was extremely carefully performed, hopefully the next one will include more precise data from areas which are non/under-reporting. I do not have any further reasons to delay the publication of this excellent work.

Many thanks to the editors for the opportunity to review and read this paper before others!

We sincerely thank the reviewer for taking the time to critique and strengthen our manuscript. We look forward to improving the precision of our estimates in the future as community-based representative surveys on snakebite envenoming continue to be produced.

Reviewer #2 (Remarks to the Author):

No further comments

We thank the reviewer for their time reviewing our manuscript.